# Potential Inhibitors of Lumpy Skin Disease’s Viral Protein (DNA Polymerase): A Combination of Bioinformatics Approaches

**DOI:** 10.3390/ani14091283

**Published:** 2024-04-24

**Authors:** Sabbir Zia, Md-Mehedi Sumon, Md-Ashiqur Ashik, Abul Basar, Sangjin Lim, Yeonsu Oh, Yungchul Park, Md-Mafizur Rahman

**Affiliations:** 1Department of Biotechnology and Genetic Engineering, Faculty of Biological Sciences, Islamic University, Kushtia 7003, Bangladesh; sabbirzia7@gmail.com (S.Z.); mdmehedi2121@gmail.com (M.-M.S.); ashik.btge@std.iu.ac.bd (M.-A.A.); abulbasar15821@gmail.com (A.B.); 2College of Forest & Environmental Sciences, Kangwon National University, Chuncheon 24341, Republic of Korea; sangjin@kangwon.ac.kr; 3College of Veterinary Medicine & Institute of Veterinary Science, Kangwon National University, Chuncheon 24341, Republic of Korea; yeonoh@kangwon.ac.kr

**Keywords:** lumpy skin disease, DNA polymerase, potential inhibitors, antiviral-drug compounds, molecular docking

## Abstract

**Simple Summary:**

Lumpy skin disease (LSD) presents a formidable challenge to livestock production worldwide, spreading rapidly among ruminant animals and causing significant economic losses. Despite vaccination efforts, the limitations of existing sheep and goat pox vaccine necessitate alternative therapeutic solutions. Using computational analysis, this study focuses on identifying inhibitors that target LSDV039, a critical protein associated with LSDV. Virtual screening identifies four potent inhibitors from a library of repurposed drugs and phytocompounds, while molecular dynamics simulations validate the stability and efficacy of those selected inhibitors. Past vaccine development attempts underscore the urgency of finding more successful treatments, while the potential of repurposing drugs and phytocompounds offers hope for combating LSD. Notably, this study highlights the versatility of computational methods in drug discovery and emphasizes the need for experimental validation in order to pave the way for novel therapeutic strategies against LSD. This research offers insight into the broader potential of computational approaches in combating infectious diseases.

**Abstract:**

Lumpy skin disease (LSD), caused by a virus within the *Poxviridae* family and *Capripoxvirus* genus, induces nodular skin lesions in cattle. This spreads through direct contact and insect vectors, significantly affecting global cattle farming. Despite the availability of vaccines, their efficacy is limited by poor prophylaxis and adverse effects. Our study aimed to identify the potential inhibitors targeting the LSDV-encoded DNA polymerase protein (gene LSDV039) for further investigation through comprehensive analysis and computational methods. Virtual screening revealed rhein and taxifolin as being potent binders among 380 phytocompounds, with respective affinities of −8.97 and −7.20 kcal/mol. Canagliflozin and tepotinib exhibited strong affinities (−9.86 and −8.86 kcal/mol) among 718 FDA-approved antiviral drugs. Simulating the molecular dynamics of canagliflozin, tepotinib, rhein, and taxifolin highlighted taxifolin’s superior stability and binding energy. Rhein displayed compactness in RMSD and RMSF, but fluctuated in Rg and SASA, while canagliflozin demonstrated stability compared to tepotinib. This study highlights the promising potential of using repurposed drugs and phytocompounds as potential LSD therapeutics. However, extensive validation through in vitro and in vivo testing and clinical trials is crucial for their practical application.

## 1. Introduction

Lumpy skin disease (LSD) is a contagious viral disease that affects ruminant animals such as cattle, water buffaloes, and giraffes, posing a significant threat across borders [1,2,3,4]. It was first reported in Africa (Zambia) in 1929 and was identified as a communicable disease in the 1940s [5]. In Bangladesh, the outbreak first emerged in the Chattogram region in July 2019 and then quickly spread throughout the entire country [6].

The virus is most likely spread mechanically by blood-sucking arthropods (flies, mosquitoes, and ticks) and, to a lesser extent, through infected animals’ bodily fluids (milk, blood, nasal secretions, and saliva) during feeding or contact [5,7,8]. Affected animals primarily exhibit clinical signs such as fever, nodular skin lesions, dramatic decreases in milk production, and weight loss [9]. The disease infects 5–45% of a herd and kills 0.5–7% of infected animals [10,11]. Consequently, it poses a significant economic threat to the livestock industry worldwide, causing abortions, male sterility, and vast production losses [12].

The effective prevention and elimination of infectious diseases relies heavily on the widespread implementation of their respective vaccines [13]. To date, very few vaccines have been able to stimulate immune responses owing to their cross-protection against LSD. Live attenuated vaccines, like Kenyan sheep and goat pox (KSGP) O-180, fail to immunize against LSDV, the while Gorgan goat pox (GTP) vaccine effectively manages clinical symptoms [14,15]. The inactivated-strain Neethling vaccine shows variable efficacy across regions [16,17,18,19]. The recombinant vaccines for Kenya sheep-1(KS-1) protect against LSDV, whereas the LSD OBP vaccine shows severe post-vaccinal reactions [20,21]. Despite these efforts, there are no DIVA (differentiating infected from vaccinated animals) or subunit vaccines available for LSD [22]. However, these vaccines provide only partial or incomplete protection against lumpy skin disease virus and exhibit post-vaccination clinical signs, poor immunogenicity, and adverse consequences [23]. Additionally, they are expensive, necessitate high dosages, and are sensitive to temperature variations, posing challenges in terms of transportation [24].

Lumpy skin disease virus (LSDV) is a member of the Capripoxvirus genus [25]. It is a double-stranded DNA virus with a genome of approximately 151 kilobase pairs (kb), encoding 156 putative genes [26]. Similar to other poxviruses, LSDV possesses several conserved genes (LSDV039, LSDV077, LSDV082, LSDV083, LSDV112, LSDV133, and LSDV139) for use in DNA replication [26]. The encoding of the DNA and RNA polymerase proteins are essential for the replication and propagation of LSDV within infected host cells [27]. Understanding the functions of the LSDV DNA polymerase can aid in the identification and development of antiviral drugs that specifically target protein [28]. However, due to the lack of a complete cure for LSD, the available treatments mainly focus on relieving the symptoms of the disease [29]. Researchers are working tirelessly to develop effective drugs to use as therapeutic agents [30]. Antiviral compounds work by suppressing the virus’s ability to cause infection and multiply within host cell [31]. They often inhibit the molecular interactions and functions needed by the virus to produce new copies of itself [32]. DNA polymerases have become extremely favourable targets for the development of antiviral drugs due to their relatively conserved characteristics [33].

Antiviral drugs are essential for combating viruses and they target either the virus itself or host cell factors. Direct antiviral-drug targets include inhibitors of virus attachment, entry, uncoating, polymerases, proteases, and others [31]. However, natural compounds with significant bioavailability and low levels of cytotoxicity are the most promising options for antiviral therapy [34]. Additionally, repurposed drugs, originally developed for other diseases, have shown efficacy against viral diseases. For example, the DNA polymerase inhibitor Zidovudine (AZT), which was the first successful repurposed drug used to treat human immunodeficiency virus (HIV) in 1987, was originally developed to treat cancer [35,36]. Recently, a drug named Remdezivir (RDV), a viral DNA synthesis inhibitor and a repurposed drug approved by the Food and Drug Administration (FDA) in 2020, was used to treat COVID-19 patients [37]. Additionally, a recent in vitro study showed that ivermectin, an antiparasitic drug, is effective against capripoxviruses [38]. Nevertheless, plant-based phytocompounds with antiviral activity are suitable natural solutions and show high efficacy in inhibiting several viral diseases, such as HIV and COVID-19 [39].

The main focus of this study was to utilize computational tools, i.e., molecular docking and molecular dynamics (MDs) simulation, to examine the effective binding interactivity and affinities of plant-based phytocompounds and repurposed antiviral drugs with LSDV DNA polymerase protein and identify the finest ligand hits. Additionally, various pharmacokinetic properties, including absorption, distribution, metabolism, toxicity, and excretion (ADMET), were evaluated in order to select the most suitable candidates. Finally, molecular dynamics simulation was employed to confirm the phytocompounds and repurposed antiviral drugs with the highest potential. We chose the compounds possessing the highest binding energy and strongest stabilizing capacity to target receptors.

## 2. Materials and Methods

### 2.1. Target Selection and Validation

In this study, target proteins were screened after an extensive literature review (Appendix A). The proteins chosen are essential for LSDV replication and the survival of LSDV [14]. Interestingly, most of these proteins share sequence similarities with sheeppox virus (SPPV) and goatpox (GTPV) [40]. The amino acid sequences of these proteins were obtained from UniProt KB (https://www.uniprot.org/ (accessed on 20 October 2023)), a widely used protein database, and their structures were predicted using AlphaFold2, an advanced protein structure prediction tool [41]. To validate the predicted structures, we used the PROCHECK [42] and ERRAT plot [43] programs. These are available from the Structural Analysis and Verification server (SAVES) (http://nihserver.mbi.ucla.edu/SAVES (accessed on 1 November 2023)), which is commonly used to assess the quality of protein structures [44]. The conserved domains and superfamilies of the selected proteins were analyzed using Chimera X software v1.7 [45], InterProScan v99.0 [46], and Pfam 36.0 (Proteins Families Database) [47]. InterProScan provides structure-based categorization, domain and homologous superfamily predictions, and other high-level results. The Pfam database stores a curated set of sequence-aligned and curated data on protein families. Additionally, the active sites of the selected proteins were predicted using the SiteMap module of Schrödinger suite 2021-2 (Schrödinger, New York, NY, USA), a suite of computational tools widely used in drug discovery and molecular modeling [48].

### 2.2. Ligand Selection

In the present study, we investigated the potential use of antiviral compounds as LSDV inhibitors. To create a dataset of active compounds, we conducted an extensive literature search of various databases and compiled a dataset comprising active phytocompounds and repurposed drug compounds. We created a library of active antiviral compounds by conducting a comprehensive search of the related literature in various databases, including PubMed, Google Scholar, Web of Science, and Scopus. We retrieved the 3D structures of active phytocompounds and drug compounds in SDF (structure–data file) format from the PubChem [49] and DrugBank [50] databases. The SDF format is a file format commonly used to represent chemical structures and associated data. The Open Babel software 2.4.1 was used to convert the SDF into the PDB format [51]. After obtaining the 3D structures of the phytocompounds, their molecular interactions, binding modes, and potential as LSDV inhibitors were investigated (Figure 1).

### 2.3. Ligand Preparation

LigPrep, a tool contained in the Schrödinger suite (Maestro 12.8) [52], was utilized for ligand preparation, which accounted for the metal-binding states, desalting, and generation of tautomers and stereoisomers. It also facilitated the generation of various possible ionization states at a targeted pH of 7.0 ± 2.0 using Epik. An OPLS4 force field was used during preparation.

### 2.4. Protein Preparation

The first step in protein processing was to extract the desired protein structure from the AlphaFold2. Subsequently, the Protein Preparation Wizard (PrepWizard), part of the Schrödinger suite (Maestro 12.8) [53], was used to perform several preprocessing steps on the protein structure. These steps involved the addition of missing hydrogen atoms, side-chain optimization, the correction of the ionization state of the protein, and the assignment of bond orders and formal charges. In addition, advanced algorithms were used to refine the protein structures. For example, they were used in the removal of water molecules and in energy minimization using the Optimized Potentials for Liquid Simulation 4 (OPLS4) force field.

### 2.5. Site Map Analysis

The selected protein was deposited on a site map, a module of the Schrödinger suite (Maestro 12.8), for use in binding-site analysis [54]. This software employs advanced algorithms to identify binding sites, evaluate the location of binding sites with a high degree of confidence, and predict the druggability of these sites.

### 2.6. Receptor Grid Generation

The interaction region between the protein and ligand was determined by generating a receptor grid using the receptor grid generation tool in Maestro 12.8 [55,56]. This grid defined the specific area surrounding the active site of the protein, and its dimensions were established based on the x-, y-, and z-coordinates. The receptor grid box had a resolution centered at coordinates of 4.2, 15.18, and 1.51 along the *x*-, *y*-, and *z*-axes, respectively.

### 2.7. Molecular Docking

#### 2.7.1. Virtual Screening and Ligand Docking

During drug development, a computational approach known as virtual screening or virtual ligand screening was used to search small-molecule libraries and identify drug-like compounds capable of binding to therapeutic targets [57]. It has been used to discover novel chemical entities in structure-based drug design. We assigned 718 antiviral compounds (Appendix A) to the virtual screening workflow (VSW) in Maestro 12.8 [58,59,60].

The high-throughput virtual screening (HTVS) mode of Glide was used during the initial step. GLIDE (grid-based ligand docking with energies), a powerful tool for drug discovery, accelerated accurate molecular docking. It employed two key scoring functions: the Emodel function efficiently differentiated between the active and inactive ligand–protein complexes of a given ligand, while the glide score effectively ranked candidate compounds, prioritizing those with strong binding affinities (actives) over those with weak interactions (inactives) [61]. The top 50% of the resulting ligands were retained for further analysis in the subsequent stage, which involved Glide Standard Precision (SP). Once again, the top-scoring 50% of the ligands were selected for use in the Glide Extra Precision (XP) modes [62].

The screened compounds were again examined in the ligand-docking module in the Glide Extra Precision (XP) mode, as it provided greater accuracy. The docking process used a flexible docking mode that automatically generated conformations for each ligand input. The G Score of the glide was examined as follows:G Score = a × vdW + b × Coul + Lipo + H-bond + Metal + BuryP + RotB + site(1)
where vdW is the Van der Waals energy, Coul is the Coulomb energy, Lipo is the lipophilic contact, HBond denotes hydrogen bonding, Metal means metal binding, BuryP indicates the penalty for buried polar groups, RotB means the penalty for freezing rotatable bonds, Site denotes polar interactions in the active site, and a = 0.065 and b = 0.130 are the coefficients of vdW and Coul, respectively, in Equation (1).

Furthermore, additional settings were configured, including the incorporation of Epik state penalties into the docking score, the execution of post-docking minimization, and the computation of the root-mean-square deviation (RMSD) with respect to the input ligand geometries. This protocol provides the top-scoring ligands in the XP description based on the glide score and glide energy. Glide calculates the energies of a wide variety of interactions between ligands and proteins, such as hydrophobic interactions, hydrogen bonds, internal energies, pi-stacking interactions, salt bridges, desolvation, and RMSDs.

#### 2.7.2. Free Energy Calculation by MM-GBSA

Following docking in Glide, both the receptor and ligands were considered using Prime MM-GBSA (molecular mechanics-generalized born surface area). MM-GBSA offers a balance between accuracy and efficiency in predicting how tightly small molecules bind to biological targets. It primarily relies on conducting molecular dynamics-based simulations of the receptor–ligand complex, being positioned between empirical scoring and a rigorous alchemical perturbation approach [63]. The Prime MM-GBSA module of the Schrodinger suite 2021-2 was employed to calculate with high accuracy the binding free energies of the protein–ligand complexes [64,65]. After selecting the receptor and ligand, Prime MM-GBSA uses the pose viewer file (pv.maegz) to show the appropriate descriptions [66]. The ligand–protein binding energy (G bind) was estimated using the following equation:ΔG bind = G complex − (G protein + G ligand)(2)
where the terms G complex, G protein, and G ligand, shown in Equation (2), represent the minimum free energies associated with the protein–ligand complex, free protein, and free ligand, respectively.

#### 2.7.3. ADMET Profiling of Novel Antiviral Compounds against LSDV

Ligands with the highest glide scores were carefully selected after virtual screening and ligand docking. To ensure that these selected ligands had the potential to be effective drug candidates, we performed a thorough evaluation using the Lipinski rule and ADMET analysis. We conducted this analysis using the QikProp module of the Schrödinger suite, specifically using Maestro 12.8 [67].

#### 2.7.4. Molecular Dynamics (MDs) Simulation

MDs simulations were performed using the Schrödinger LLC Desmond software v.2022-3 for a duration of 100 nanoseconds [68]. Prior to MDs simulations, a crucial docking step was performed to predict the static binding position of the ligand at the active site [69]. The MDs simulations employed Newton’s classical equation of motion to simulate atomic movements over time and predict the ligand-binding status in a physiologically relevant environment [70]. The ligand–receptor complex was prepared using Maestro’s Protein Preparation Wizard, which involved the optimization, minimization, and filling of missing residues, if necessary. The system was created using a built-in tool.

The MDs simulations were performed with the TIP3P (Intermolecular Interaction Potential 3 Points Transferable) solvent model in an orthorhombic box, which allowed for a 10 Å buffer region between protein atoms and box sides while maintaining a temperature of 300 K and pressure of 1 atm. We employed an Optimized Potentials for Liquid Simulation (OPLS 2005) force field to describe the interactions within the system [71].

To mimic physiological conditions, counterions, and 0.15 M of sodium chloride were added to neutralize the total charge of the system. The models were loosened prior to the actual simulation. Trajectories were stored and inspected at regular intervals of 100 ps for further analysis. Furthermore, the trajectories obtained from the simulations were utilized in the Veusz software v. 3.6.2 (https://veusz.github.io/ (accessed on 20 November 2023)) to analyze structural insights and stability through root-mean-square deviation (RMSD), root-mean-square fluctuation (RMSF), the radius of gyration (Rg), and solvent-accessible surface area (SASA).

## 3. Results

### 3.1. LSDV DNA Polymerase Protein Has a Groove-like Active Site

The identification of the key proteins associated with diseases is a fundamental step in understanding their underlying mechanisms and developing targeted therapeutic interventions to treat them [72]. In this study, we focused on the ability of protein to replicate DNA during the LSDV life cycle. Based on their DNA replication ability, 12 of the 156 putative genes encoded by the LSDV genome were identified as potentially playing roles in viral DNA replication (Appendix A) [14,73]. After performing the previously mentioned structural analysis and verification, we only selected the LSDV039 gene that encoded DNA polymerase enzyme. Its principal function is to faithfully duplicate the genome, ensuring that genetic information is preserved and passed on from generation to generation [14,74]. That is why this study selected the DNA polymerase, which was the logical target for drug development against LSDV.

We performed analysis using the Chimera X software and revealed that the DNA polymerase belongs to the DNA-directed DNA polymerase family B. It has four functional domains: the DNA polymerase B exonuclease N-terminal (residues 1–22), DNA-directed DNA polymerase family B exonuclease (residues 64–349), DNA-directed DNA polymerase family B viral insert (residues 350–481), and DNA-directed DNA polymerase family B multifunctional domain (residues 493–989) (Figure 2). Additionally, the findings from the InterProScan and Pfam analyses supported the previously mentioned information. According to InterProScan, the conservation of the site within the 2–515 residues and 520–1003 residues of amino acid residues indicates that the substance being examined belonged to the ribonuclease H-like superfamily and the DNA/RNA polymerase superfamily.

We carried out a DNA polymerase protein structure prediction using AlphaFold 2, and the analysis of its binding site through the site map revealed that the active site’s structure closely resembled a groove. Additionally, a site map with a high DScore of 1.048 was selected as the active site. Proteins with Dscores between 0.7 and 0.8 are moderately druggable, while those with Dscores greater than 1.0 are considered very druggable [75]. Active-site residues were used for receptor grid generation, followed by molecular docking (Table 1, Figure 3).

### 3.2. Structure Validation

To assess the quality of the 3D model, a Ramachandran plot was constructed using the PROCHECK software v2.3 (Figure 4). Ramachandran plot analysis of the predicted model revealed that 91% of the residues were located in the most favorable regions, whereas 8.0% were in the allowed regions. This confirmed that the predicted model was of excellent quality. A high-quality model is generally considered to have a value of >50. According to the ERRAT server, the current 3D model’s overall quality factor was 88.8 (Figure 5). The overall structural validation results are presented in Table 2.

### 3.3. Binding Profile Analysis of Bonded Interactions

Potential antiviral compounds were screened from a total of 1098 compounds using a virtual screening workflow (VSW). This was followed by ligand docking in Glide Extra Precision (XP) modes. Among the compounds selected, 380 were phytocompounds and 718 were antiviral drugs. VSW was utilized to separate 160 conformers from the 1098 samples, and performing further interaction analyses of the protein–ligand complex yielded the 10 best compounds using the results of ligand docking in Glide Extra Precision (XP) modes. Compounds with higher Glide Gscores (Gscore is a specific scoring function that is used to quantify the strength of the binding interaction between the ligand and protein) and binding interactions were evaluated as being the best conformers (Appendix A).

The ligand docking performed using Maestro (Schrodinger) showed that canagliflozin had the best Glide Gscore of −9.86 kcal/mol, with a binding affinity of −45.68 kcal/mol. The compound known as tepotinib had a Glide Gscore −8.86 kcal/mol and its binding energy was −47.99 kcal/mol. Rhein and taxifolin obtained Glide Gscores −8.97 kcal/mol and −7.20 kcal/mol, respectively, showing binding affinities −44.72 kcal/mol and −44.48 kcal/mol (Table 3). The positive control was also depicted with a Glide Gscore of −5.63 kcal/mol, showing a binding affinity of −26.28 kcal/mol.

The interaction of amino acid residues with selected ligands caused canagliflozin to establish two non-covalent bonds with the amino acid residues Lys 483 and Asn 659 (Table 4, Figure 6C). Canagliflozin formed two hydrogen bonds with Asn 659, with bond distances of 1.80 and 1.73 Å, respectively. In addition, a salt bridge was formed between Lys 483 and canagliflozin, with a bond distance of 2.61 Å. Tepotinib formed two non-covalent interactions. These involved the formation of hydrogen bonds with Glu 339 residues, with a bond distance of 1.93 Å, and the formation of a salt bridge with Glu 399, with a bond distance of 2.94 Å. Moreover, it also established four pi interactions, in which Tyr 477, Phe 499, and Tyr 663 amino acid residues were involved in pi–pi stacking with bond distances of 5.49, 4.89, and 5.38 Å, respectively, whereas pi–cation was only formed with amino acid residue Lys 483, having a bond distance of 4.01 Å (Figure 6D). Rhein formed two non-covalent interactions involving two hydrogen bonds with Val 498 and Thr 664 residues, having a bonding distance of 2.12 and 2.09 Å, respectively (Figure 6E). It also established a salt bridge with Lys 337 with a bond distance of 2.71 Å. Taxifolin was involved in forming five non-covalent hydrogen bonds with amino acid residues Ser 338, Glu 339, Lys 483, Ser 656, and Thr 664, having a bond distance of 2.10, 1.78, 2.00, 2.02, and 2.05 Å, respectively. It was also involved in a pi interaction, specifically between pi and cation, with Lys 483, having a bond distance of 3.90 Å (Figure 6F). In the case of the positive control, it formed three non-covalent hydrogen bonds, with Asp 337, Ser 338, and Glu 339 residues, where the bond distance between them was 1.83 Å, 2.16, and 1.55, respectively (Figure 6).

### 3.4. Binding Free Energy Calculations for Non-Bonded Interactions

Non-bonded interactions are more abundant than bonded interactions during protein–ligand docking. One of the most significant non-bonded interactions is the Van der Waals energy interaction. Coulomb energy is also one of the major forms of non-bonded energy. In addition, lipophilic energy, which is an important non-bonded interaction, is a ligand–receptor complex. Binding free energy calculations, performed using the Prime MMGBSA module, reveal that canagliflozin formed a Van der Waals interaction with an energy of −42.95 kcal/mol. Tepotinib formed Van der Waals interactions with energy of −58.20 kcal/mol. Rhein and taxifolin are involved in Van der Waals interactions, with energies of −34.55 kcal/mol and −24 kcal/mol, respectively. The positive control formed anenergy with −63.90 kcal/mol. Other forms of non-bonded interactions and their energies are listed in Table 5.

To determine the non-bonded interaction energy in the protein–ligand complex, the following equation was employed: MMGBSA ∆G Bind (NS) = MMGBSA ∆G Bind − Rec Strain − Lig Strain.

### 3.5. ADMET Profiling of Novel Antiviral Compounds against LSDV

We identified the top ten potential antiviral compounds based on their high Glide Gscore, indicating their promising binding affinity to viral targets. These compounds were subjected to thorough ADMET (absorption, distribution, metabolism, excretion, and toxicity) analyses using the QikProp module, which assesses crucial drug properties. The selected compounds exhibited favorable physicochemical and ADMET properties, making them excellent candidates for antiviral-drug studies (Table 6).

### 3.6. Molecular Dynamics (MDs) Simulation

MDs simulations were performed to study the stability and behavior of the protein–ligand complexes over time. In this study, we examined the control complex and the four newly chosen compounds in order to understand their characteristics through a range of analyses, including RMSD, RMSF, Rg, and SASA. As shown in Figure 7a, the selected compound Rhein showed the lowest RMSD values compared to the positive control and the other selected compounds. A lower RMSD value indicates a higher level of system compactness [76]. The average RMSD values of ivermectin B1a, canagliflozin, tepotinib, rhein and taxifolin were 0.88 Å, 0.88 Å, 1.44 Å, 0.33 Å, and 0.51 Å, respectively (Table 7). Canagliflozin exhibited a stable conformation from 15 to 63 ns. Tepotinib showed a stable conformation from 1 to 77 ns, after which fluctuations were observed from 78 to 100 ns. Rhein exhibited structural stability from 30 to 63 ns, whereas the remainder fluctuated. Taxifolin showed an initial fluctuation until 10 ns, after which a specific stabilization could be observed from 10 ns up to 89 ns; again, fluctuation occurred from 90 to 91 ns, and the remainder of the values showed structural stability. The control drug ivermectin B1a showed stability and compactness at 16–36 ns and 44–100 ns, respectively.

Additionally, RMSF analysis was employed to examine the average atomic displacement from the mean positions in the residues during molecular dynamics simulation (Figure 7b). The canagliflozin showed the highest average RMSF value (1.69 Å), while the tepotinib showed the lowest RMSF value (1.26) among all systems. Higher RMSF values indicate that the system has greater flexibility during the molecular dynamics simulations [66]. The average RMSF value of Rhein (1.62 Å) was quite similar to the RMSF value of taxifolin (1.63 Å). However, ivermectin B1a (control) showed average RMSF values of 1.58 Å (Table 7).

The radius of gyration (Rg) was used to measure the compactness of the system throughout the simulation (Figure 7c). Herein, the positive-control ivermectin B1a showed the lowest Rg value of 32.72 Å among all the systems. In contrast, canagliflozin showed an average effect.

The Rg value (32.97 Å) was quite similar to those of tepotinib (32.99 Å) and taxifolin (32.89 Å). However, rhein showed a slightly higher Rg value of 33.18 Å (Table 7). Figure 7d illustrates the SASA analysis for all systems. The SASA value provides insights into the compactness of the systems, with lower values indicating higher compactness, and higher values suggesting openness [73]. The selected complexes had average SASA values of 47,186.72, 47,677.70, 47,724.08, 48,127.33, and 47,439.90 Å for ivermectin B1a, canagliflozin, tepotinib, rhein, and taxifolin, respectively, as indicated in Figure 7. Among all systems, the positive control, ivermectin B1a, displayed the lowest average SASA value of 47186.72 Å. The average SASA values of canagliflozin (47,677.70 Å), tepotinib (47,724.08 Å), and taxifolin (47,439.90 Å) were quite similar, while rhein showed the highest SASA values among all systems, standing at the slightly higher value of 48,127.33 Å.

## 4. Discussion

The repurposed drugs and phytocompounds proved their ability as alternative treatment options for lumpy skin disease [77,78]. This research aimed to predict the binding affinity of repurposed drugs and phytocompounds to LSDV DNA polymerase using in silico molecular docking and simulations. The top four compounds, two repurpose drug compounds (canagliflozin and tepotinib) and two phytocompounds (rhein and taxifolin), were identified as highly promising LSDV DNA polymerase protein inhibitors.

This study revealed that all selected compounds formed different interactions with specific active-site residues of the LSDV DNA polymerase protein. The formation of hydrogen bonds between the ligand and protein is highly selective and specific [79]. This depends on the spatial arrangement of the atoms in relation to both the binding site of the protein and the ligand. The complementarity of these structures allows precise interactions to occur. The presence of hydrogen bonds stabilizes the ligand–protein complex, enhancing binding specificity [80]. Our results showed that each of the selected compounds formed hydrogen bonds with specific active-site residues of the LSDV DNA polymerase protein (Table 4), maintaining a specific distance. Taxifolin formed the highest number of hydrogen bonds. Moreover, various non-covalent interactions such as pi–pi interactions, salt bridges, and cation–pi bond interactions were observed, contributing to the stability of ligand–protein complexes [81,82,83]. Furthermore, the drug-like properties of the tested compounds satisfied Lipinski’s rule of five, indicating their potential as drug candidates. Pharmacokinetics and toxicology analyses further supported their viability [84].

Molecular simulations are a powerful approach to understanding the stability and dynamics of protein–ligand complexes [85,86]. In this study, rhein exhibited the highest stability among the tested compounds, followed by taxifolin, tepotinib, and canagliflozin. The stability and dynamics of the compounds in protein–ligand complexes were studied. Through dynamic simulation spanning 100 ns, it was determined that the phytocompound rhein showed the highest stability in RMSD and RMSF. It maintained structural stability from 30 to 63 ns, with slight fluctuations compared to the positive control. Rhein exhibited less compactness and less exposure to the surface area, as indicated by its Rg and SASA values, respectively. Similarly, the other tested compounds, including taxifolin, tepotinib, and canagliflozin exhibited mixed results across RMSD, RMSF, Rg, and SASA. While taxifolin and canagliflozin exhibited fluctuations, they maintained some level of stability throughout the simulation. Tepotinib initially displayed compactness, but became unstable towards the end of the simulation. Similar dynamic simulation techniques were used to screen drugs for COVID-19 [87].

The advancement of technology has revolutionized the field of structure-based drug discovery, allowing for the rapid virtual screening of large compound libraries and molecular dynamics simulations [88]. For example, a structure-based nelfinavir drug was discovered in the 1990s for the treatment of human immunodeficiency virus (HIV) infection [89]. Canagliflozin and tepotinib are repurposed drugs, which means they have already been approved for use in treating specific diseases, and currently, there is interest in exploring their potential use for different diseases, including LSD. Canagliflozin (Invokana) is primarily used for the treatment of type 2 diabetes [90,91,92]. Canagliflozin exhibits a high capacity to bind plasma proteins (99%), mainly albumin. Their effectiveness in preventing viral replication will need more investigation. Tepotinib belongs to a class of medications known as kinase inhibitors. It works by blocking the action of an abnormal protein that signals cancer cells to multiply [93,94,95]. As with an in silico approach, tepotinib was screened as a Mpox virus inhibitor based on binding free energy and conformational behavior analysis [96]. Similarly, this study identified two phytochemicals, rhein and taxifolin, as promising compounds with drug-like properties for inhibiting the macro domain of the chikungunya virus [97,98]. These compounds were investigated for their binding affinities to the active sites of the DNA polymerase protein LSDV039, crucial for antiviral-drug development. Additionally, two repurposed drugs, canagliflozin and tepotinib, were examined. Post-MM/GBSA analysis indicated that the complexes had improved binding free energy at 100 ns, suggesting effective interaction with target proteins, potentially inhibiting their activity and interfering with viral replication. Previous studies have shown that phytochemicals derived from *Moringa olifera* can inhibit the DNA polymerase of the monkeypox virus, indicating their potential as antiviral agents. However, to develop these selected compounds as LSD-specific antivirals, it is crucial to understand their behavior in natural hosts, confirm their ability to inhibit replication and provide antiviral prophylaxis. Notably, the identified compounds (canagliflozin, tepotinib, rhein, and taxifolin) are publicly available, which facilitates their rapid utilization and will enable further research. Further experimental studies are necessary to validate their efficacy as novel compounds against LSDV.

An in vitro study found that ivermectin (a repurposed drug, mainly used as anti-parasite medication) demonstrated significant inhibitory effects on viral replication and on the attachment and penetration stages of the LSDV virus. Specifically, the results showed that ivermectin reduced the viral replication of LSDV and SSPV by 99.82 and 99.87% at the replication stage, respectively. It also exhibited inhibitory effects of 68.38 and 25.01% at the attachment stage and 57.83 and 0.0% at the penetration stage [38]. Additionally, a recent study found that phytocompounds of *Moringa olifera* inhibited the DNA polymerase of monkeypox virus (MPXV) [86]. DNA polymerase is responsible for DNA replication. Due to its crucial role in viral replication and lifecycle, it is a prime target for antiviral-drug development [28]. Therefore, the inhibition of this enzyme may cause viral replication to stop and can also reduce viral proliferation [99]. Moreover, the limitations of this study are highlighted. Firstly, none of the proposed treatments have been experimentally validated, although similar in silico approaches have demonstrated success in drug discovery, such as the identification of remdesivir for Ebola virus treatment through similar screening methods [100]. Similar approaches have been used to identify drugs that have progressed to experimental validation [101]. The present study’s finding may be that their interactions are certain rather than predictions. The second important limitation of the current study is the selection of ivermectin as the “positive control”. While ivermectin has been shown to have the capacity to inhibit the lifecycle of LSDV, this effect may not only be exercised through the inhibition of DNA polymerase, but also through penetration and attachment stages [38]. The exact mechanism of the inhibition behind nodular dermatitis viruses like LSDV remains unclear. Additionally, recent studies on bovine herpesvirus 1 have elucidated some of the underpinning mechanisms of how ivermectin inhibits this virus. These studies suggest that ivermectin interferes with the transportation of viral proteins (including DNA pol) into the nucleus [102]. Of course, this mode of action would not be effective against poxviruses. However, published studies have identified potential inhibitors of monkey poxvirus DNA polymerase, and perhaps one of these might be a more appropriate control compound [103].

Overall, LSD continues to pose significant challenges to the livestock industry, and more comprehensive control measures, including improved vaccination coverage and alternative treatment options, are required to mitigate its impact. Based on in silico screening of natural compounds, that is, rhein and taxifolin, and repurposed drugs, that is, canagliflozin and tepotinib, combinatorial docking, molecular dynamic simulation, drug-likeliness analysis, and all other experimental data were used.

## 5. Conclusions

This study emphasized the importance of developing alternative treatment options for LSD and highlighted the potential of repurposed drugs and phytochemicals as antiviral agents. These findings suggest that canagliflozin, tepotinib, rhein and taxifolin have strong binding affinities for important viral proteins, exhibit favorable dynamics and hydrogen-bonding patterns, and possess favorable drug-like properties. This study serves as a foundation for future research and for the design of specific drugs targeting LSD. Experimental studies are required to confirm the efficacy of these phytocompounds as potential therapeutic agents against viruses.

## Figures and Tables

**Figure 1 animals-14-01283-f001:**
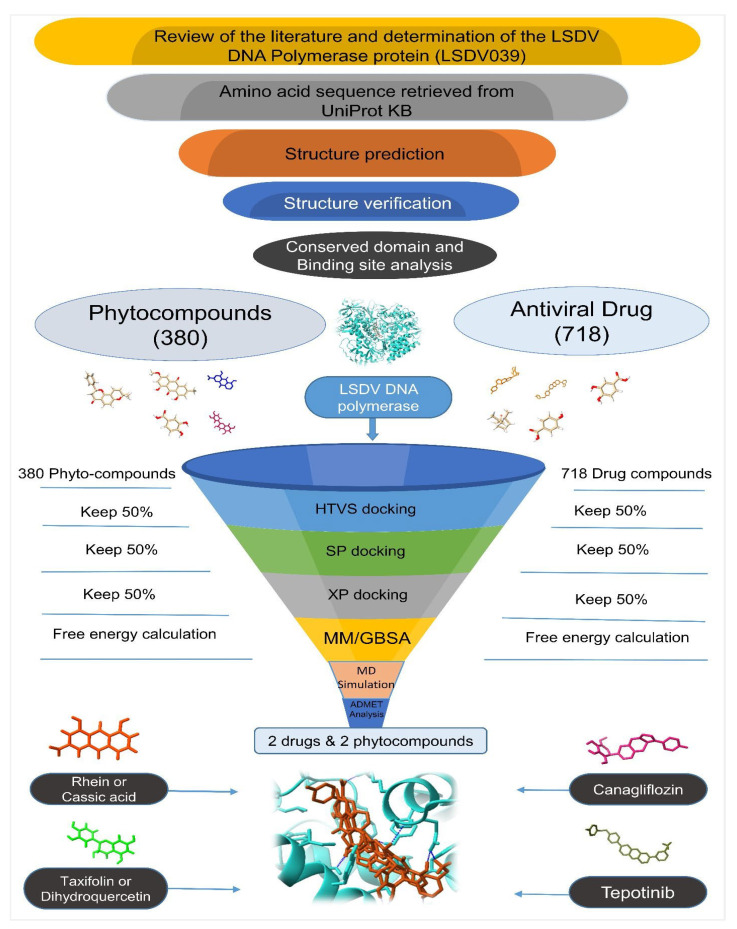
A schematic representation of structural base virtual screening and a molecular docking study using repurposed drugs and phytocompounds.

**Figure 2 animals-14-01283-f002:**
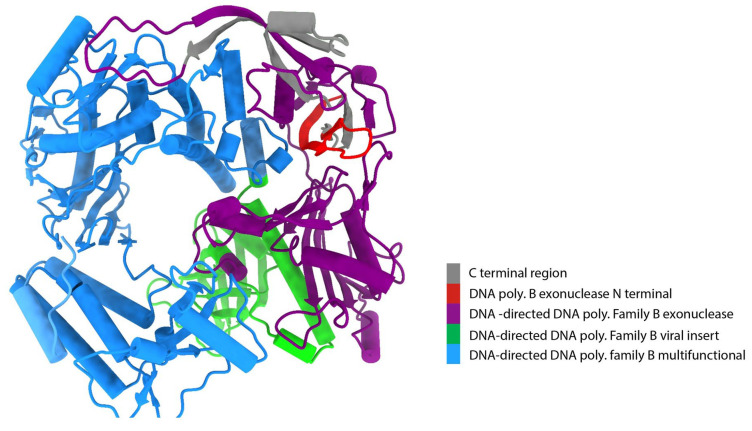
Conserved domains in DNA polymerase uncovered using Chimera X. Gray color indicates the carboxy terminal of DNA polymerase. The four main functional domains of DNA-directed DNA polymerase family B represent four distinct colors: orange color indicates DNA polymerase B exonuclease N-terminal, purple color indicates DNA-directed DNA polymerase family B exonuclease, green color indicates DNA-directed DNA polymerase family B viral insert, and blue color indicates DNA-directed DNA polymerase family B multifunctional domain.

**Figure 3 animals-14-01283-f003:**
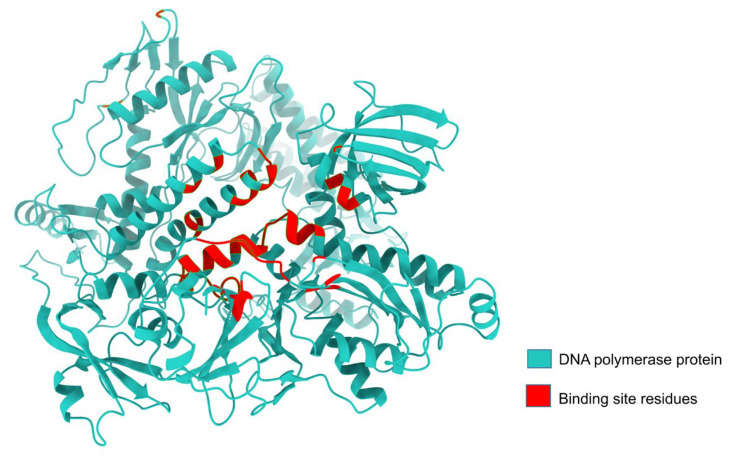
DNA polymerase protein binding site residues. The red color denotes the binding site residue, and the light sea green color represents the non-binding residues.

**Figure 4 animals-14-01283-f004:**
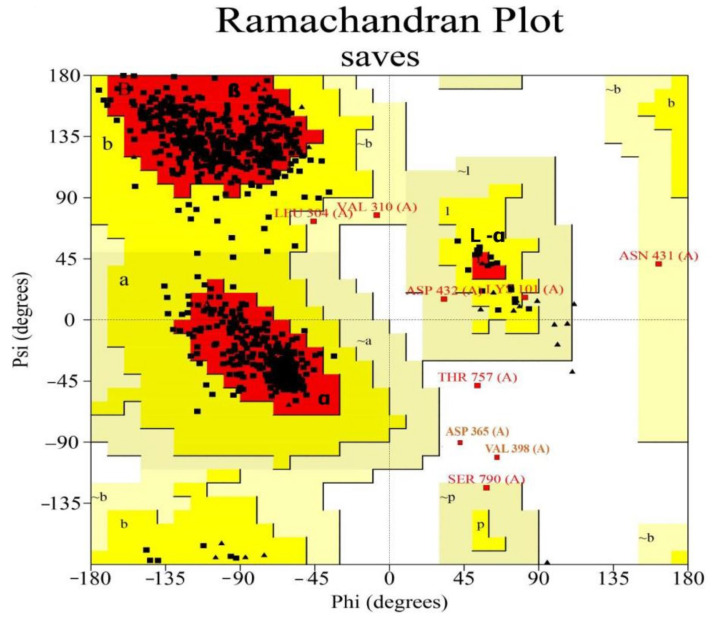
Validation of the DNA polymerase protein via Ramachandran plot. The plot presents the dihedral angles Psi and Phi of amino acid residues, where white areas indicate regions that are sterically disallowed for most amino acids, except for glycine residues. Conversely, red areas represent allowed conformations, while the yellow region denotes the outer limit of permissible conformations. Each black-and red-square/triangle on the plot represents amino acids, including the black triangle indicating glycine residue, the black square indicating non-glycine residues, and the red square residue indicating proline residue. Within the red region of the plot, the first quadrant signifies a left-handed alpha-helix, the second quadrant indicates a beta-sheet structure, and the third quadrant denotes a right-handed alpha-helix. Several amino acids overlap with each other, specifically, Leu 304 with Val 310; Asp 432 with Lys 101; and Asp 365 with Val 398. The binding amino acids in LSDV-encoded DNA polymerase gene (LSDV039) were provided in Table 1. [A, B, L] indicate residues in most favored regions; [a, b, l, p] are additional allowed regions; and [~a, ~b, ~l, ~p] are generously allowed regions.

**Figure 5 animals-14-01283-f005:**
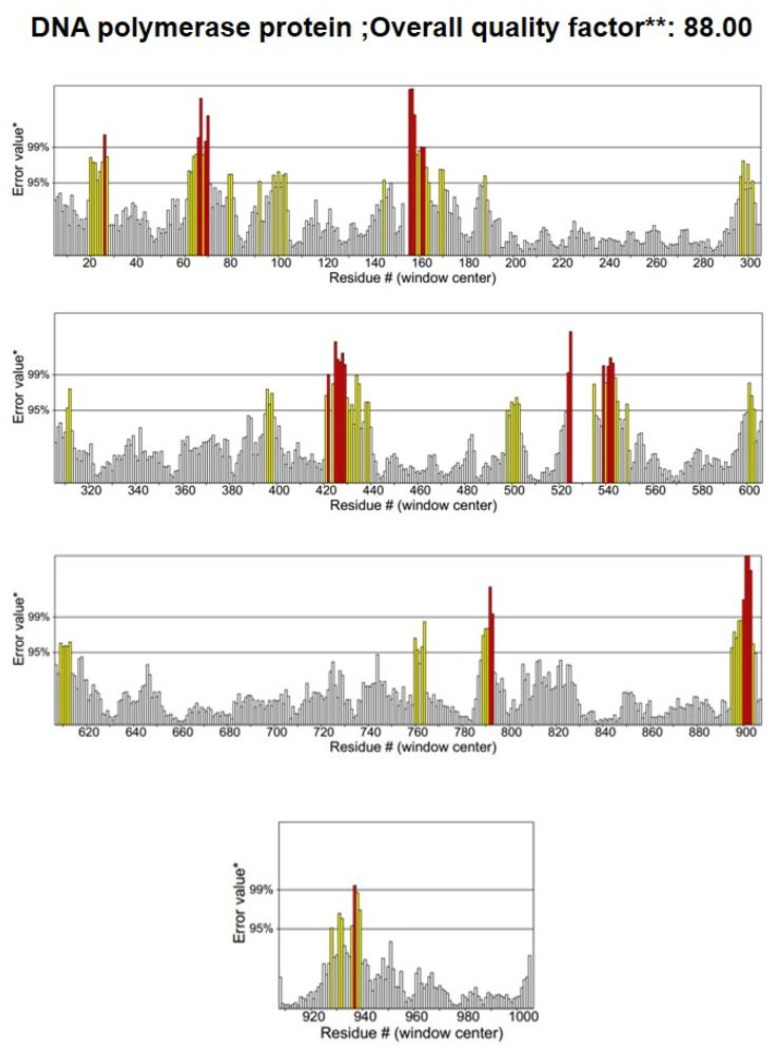
Structural validation using ERRAT2. The red bars indicate the misfolded region, yellow bars indicate the error region, and grey bars represent the region with a lower error rate. * on the error axis, two lines are drawn to indicate the confidence with which it is possible to reject regions that exceed that error value, ** expressed as the percentage of the protein for which the calculated error value falls below the 95% rejection limit. Good high resolution structures generally produce values around 95% or higher. For lower resolutions (2.5 to 3A) the avaerage overall quality factor is around 91%.

**Figure 6 animals-14-01283-f006:**
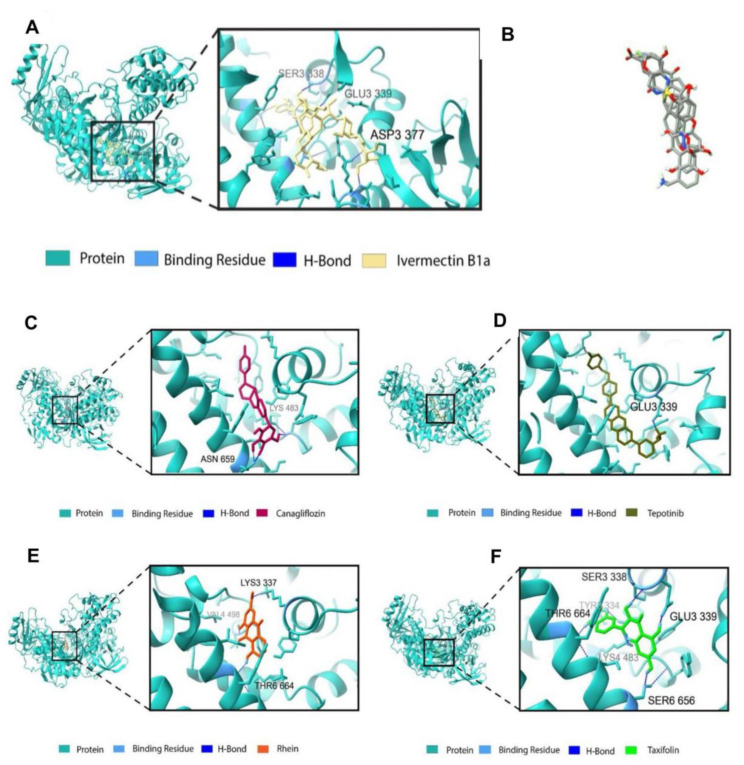
(**A**) Docking interactions of ivermectin B1a (positive control) with DNA polymerase protein. (**B**) Docking pose alignment of two drugs (canagliflozin and tepotinib) and two phytocompounds (rhein and taxifolin). Docking interactions of (**C**) canagliflozin, (**D**) tepotinib, (**E**) rhein (cassic acid), and (**F**) taxifolin (Dihydroquercetin) with DNA polymerase protein.

**Figure 7 animals-14-01283-f007:**
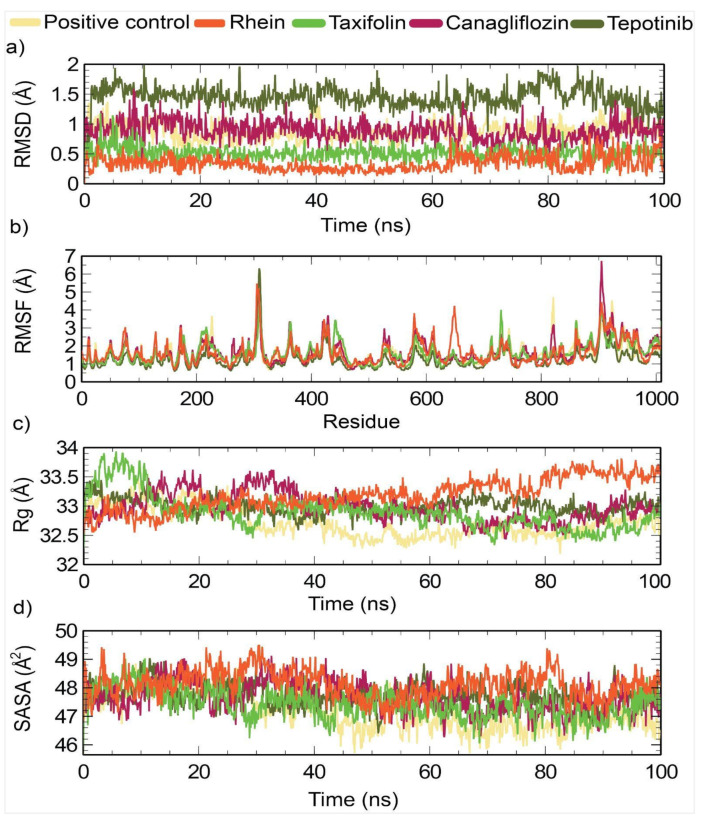
Analysis of molecular dynamic simulation for 100 (ns), applied to DNA-polymerase, where (**a**) RMSD (root-mean-square deviation); (**b**) RMSF (root-mean-square fluctuations); (**c**) Rg (radius of gyration); and (**d**) SASA (solvent-accessible surface area).

**Table 1 animals-14-01283-t001:** Binding site residues used as input for receptor grid generation during docking.

Protein Name	Binding Site Residues
Lumpy skin disease virus-encoded DNA polymerase (LSDV 039)	Lys 4, Glu 122, Gly 123, Cys 124, Arg 155, Phe156, Asn 157, Ile 158, Asn 159, Arg 160, Tyr 162, Phe 164, Ile 191, Asn 195, Leu 304, Phe 329, Thr 333, Tyr 334, Lys 337, Ser 338, Glu 339, Lys 340, Asn 352, Ala 353, Phe 354, Ser 355, Cys 356, Asn 374, Ile 379, Gly 380, Lys 381, Ile 382, Ser 383, Ser 384, Phe 385, Glu 387, Val 388, Asp 412, Tyr 473, Trp 475, Asn 476, Tyr 477, Tyr 478, Gly 479, Ile 480, Glu 481, Thr 482, Lys 483, Asp 485, Ala 486, Gly 487, Phe 489, Tyr 491, Val 498, Phe 499, Glu 500, Tyr 501, Arg 502, Ala 503, Leu 506, Tyr642, Tyr646, Leu 651, Ser 652, Thr 653, Lys 655, Ser 656, Ile 657, Tyr 658, Asn 659, Ser 660, Met 661, Glu 662, Tyr 663, Thr 664, Tyr 665, Ile 667, Ile 668, Ser 671

**Table 2 animals-14-01283-t002:** Protein structure validation through PROCHECK (Ramachandran plot) and Errat plot.

Gene	Uniprot ID	Site Map Analysis	Errat Value	Ramachandran Plot			
				Most Favored Regions	Additional Allowed Regions	Generously Allowed Regions	Disallowed Region
LSDV039	Q91MW8	1.048	88.8	91%	8.0%	0.6%	0.3%

**Table 3 animals-14-01283-t003:** Energies for potential inhibitors of LSDV DNA polymerase.

Compound Type	Compound ID	Name	Glide Gscore (kcal/mol)	Glide Emodel (kcal/mol)	MMGBSA dGbind (kcal/mol)
Positive control Antiviral drugs	CID6321424	Ivermectin B1a	−5.63	−67.89	−26.28
	DB08907	Canagliflozin	−9.86	−69.37	−45.68
	DB15133	Tepotinib	−8.86	−88.69	−47.99
Phytocompounds	CID 10168	Rhein	−8.97	−43.08	−44.72
	CID 439533	Taxifolin	−7.20	−51.87	−44.48

**Table 4 animals-14-01283-t004:** Interacting residues of protein with selected ligands with bond type.

Compound Type	Compound ID	Name	Residues in Interaction	Bond Distance (Å)	Bond Type
Positive control	CID6321424	Ivermectin B1a	Asp 337 Ser 338 Glu 339	1.83 2.16 1.55	H-bond H-bond H-bond
Antiviral drugs	DB08907	Canagliflozin	Lys 483 Asn 659 Asn 659	2.61 1.80 1.73	Salt bridge H-bond H-bond
	DB15133	Tepotinib	Glu 339 Glu 339 Tyr 477 Lys 483 Phe 499 Tyr 663	1.93 2.94 5.49 4.01 4.89 5.38	H-bond Salt bridge Pi–pi stacking Pi–cation Pi–pi stacking Pi–pi stacking
Phytocompounds	CID 10168	Rhein	Lys 337 Val 498 Thr 664	2.71 2.12 2.09	Salt bridge H-bond H-bond
	CID 439533	Taxifolin	Ser 338 Glu 339 Lys 483 Lys 483 Ser 656 Thr 664	2.10 1.78 2.00 3.90 2.02 2.05	H-bond H-bond H-bond Pi–cation H-bond H-bond

**Table 5 animals-14-01283-t005:** Binding free energy of 4 selected ligands against LSDV DNA Polymerase.

Compound Type	Compound ID	Name	∆G Bind (NS)	∆G Coulomb (NS)	∆G Lipo (NS)	∆G vdW (NS)
Positive control Antiviral drugs	CID6321424	Ivermectin B1a	−39.70	−15.52	−17.01	−63.90
	DB08907	Canagliflozin	−51.44	−46.81	−25.21	−42.95
	DB15133	Tepotinib	−54.06	50.09	−23.28	−58.20
Phytocompounds	CID 10168	Rhein	−38.20	−77.35	−13.22	−34.55
	CID 439533	Taxifolin	−47.80	−29.02	−13.33	−24.43

∆G bind (NS) *=* binding/interaction energy without receptor and ligand strains. ∆G covalent (NS) *=* covalent binding energy without stains. ∆G coulomb (NS) = Coulomb energy without strains. ∆G Lipo (NS) = lipophilic energy without stains. ∆G vdW (NS) = Van der Waals energy without strains.

**Table 6 animals-14-01283-t006:** ADME analysis and pharmacological parameter of hits using Qikprop.

Compound Type	Name	#Star ^1^	Molecular Weight ^2^	SASA ^3^	FISA ^4^	QPlogPo/w ^5^	QPlogS ^6^	QPlogBB ^7^	QPlog HERG ^8^	QPlogKp ^9^	Percent Human Oral Absorption ^10^	Rule of Five ^11^	Rule of Tree ^12^
Positive control	Ivermectin B1a	11	875.10	1232.66	137.09	6.398	−8.09	−2.28	−5.98	−2.11	73.19	3	2
Antiviral drugs	Canagliflozin	4	438.47	693.64	174.33	4.01	−6.14	−1.35	−6.41	−2.97	92.33	0	2
	Tepotinib	1	494.60	881.97	94.45	4.59	−6.75	−0.55	−8.58	−3.17	100.00	0	2
Phytocompounds	Rhein	0	284.23	478.12	271.67	0.98	−2.66	−1.97	−2.69	−5.51	47.41	0	1
	Taxifolin	0	304.26	518.52	276.85	0.11	−2.73	−2.27	−4.928	−5.38	52.10	0	0

^1^ #Star—indicates values that fall outside the 95% range of similar values seen in other drugs. Many stars suggest that a molecule is less drug-like than molecules with few stars (the recommended range is 0–5). ^2^ Molecular weight (range 130.0 to 725.0). ^3^ SASA—measure the total solvent accessible surface area (recommended value 300.0 to 1000.0). ^4^ FISA—hydrophilic components of SASA (recommended value 7.0 of 330.0). ^5^ QPlogPo/w—predicts octanol/water partition coefficient (recommended value −2.0 to −6.5). ^6^ QPlogS—predicted aqueous solubility (recommended value –6.5 to 0.5). ^7^ QPlogBB—predicted brain/blood partition coefficient (recommended value –3.0 1.2). ^8^ QPlogHERG—predicted IC50 value for blockage of HERG K+ channels (recommended value, concern below –5). ^9^ QPlogKp—predicted skin permeability, log Kp (recommended value –8.0 to −1.0). ^10^ Percent Human Oral Absorption—predicted human oral absorption on a 0–100% scale (recommended value, >80% is high and <25% is poor). ^11^ Rule of Five—Lipinski’s rule of five (the maximum accepted value is four). ^12^ Rule of Three—Jorgensen’s rule of three (the maximum accepted value is 3).

**Table 7 animals-14-01283-t007:** The average mean value of a molecular dynamics (MDs) trajectory.

System	RMSD (Å)	RMSF (Å)	Rg (Å)	SASA (Å)
Ivermectin B1a	0.88	1.58	32.72	47,186.72
Canagliflozin	0.88	1.69	32.97	47,677.70
Tepotinib	1.44	1.26	32.99	47,724.08
Rhein	0.33	1.62	33.18	48,127.33
Taxifolin	0.51	1.63	32.89	47,439.90

## Data Availability

The data supporting the findings of the research are included in this article, and additional data are available in the Appendix A. Furthermore, the corresponding authors are ready to provide additional details upon request.

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
