# Peer review of "Potential Inhibitors of Lumpy Skin Disease’s Viral Protein (DNA Polymerase): A Combination of Bioinformatics Approaches"

_animals, 2024, doi:10.3390/ani14091283_

Round 1
Reviewer 1 Report
Comments and Suggestions for Authors
The study aimed to identify inhibitors of lumpy skin disease virus that could act by means of inhibition of replication of the virus.
In general, I have no major concerns about the manuscript, although some improvements can be effected in order to produce a better revised version. These are noted below.
- The manuscript does not mention clearly and explicitly the main question addressed by the research. The authors must rectify this by adding a separate paragraph in the Introduction.
- The authors do not provide adequate evidence regarding the novelty of the manuscript. In the revised version, the authors must elaborate in the Discussion about the novelty and the originality of the manuscript. They do not make the reader understand what specific gap in the field the paper address. This should be made clear in both the Introduction and in the Discussion.
- The authors do not elaborate on the value added by their manuscript in comparison to other published material. The authors must add a separate sub-section in the Discussion to present the additions in the literature brought forward by their study.
- With regard to methodologies, the authors must specific justification for implementing the equations used in their study. Not all readers may be familiar with such complex equations, therefore the authors must explain in details why they used these equations and also they must justify their use. The authors should use confirmed negative control and confirmed positive controls in all their laboratory work and also they must present data about validation of their findings versus the confirmed negative and positive control material.
- The concluding remarks about canagliflozin and tepotinib have strong binding affinity with the viral proteins have not been fully documented in the manuscript and thus are not consistent with the results and the arguments in the Discussion. The authors should tone down their conclusions. Also the comment about the future performance of experimental studies should be fully justified, as it is not covered well in the Discussion.
- Some recently published (January and February 2024) references are missing and can be added to improve the Discussion.
- Comments on figures: the visualisation of the manuscript is above average, but the authors must use similar colours to depict the same structure throughout the manuscript. -For the two figures of the protein structure, please add a new presentation of the structure from a different viewpoint to allow better understanding of the structure by readers.
-The authors should make clear that the study refers to an in silico project, as this does not become clear at the moment.
-The Introduction should be shorten, as it includes some well-established information.
-The equation for the examination of the glide can be inserted within a box, for better presentation.
-Same for the equation for the energy for the protein ligand.
-For the two figures of the protein structure, please add a new presentation of the structure from a different viewpoint to allow better understanding of the structure by readers.
-The Discussion should be divided into three sub-sections for better flow of the reading.
-Some recent references are missing and can be added to improve the Discussion.
-Please add a subsection in Discussion emphasizing the clinical significance of the findings.
Overall: Acceptance after making the above changes.
Author Response
Reviewer 1
Comments and Suggestions for Authors
The study aimed to identify inhibitors of lumpy skin disease virus that could act by means of inhibition of replication of the virus.
In general, I have no major concerns about the manuscript, although some improvements can be effected in order to produce a better revised version. These are noted below.
Response: Thanks for reviewer positive impression, valuable suggestions, proposed revisions, which make the present manuscript would be better for accepting the Journal.
- The manuscript does not mention clearly and explicitly the main question addressed by the research. The authors must rectify this by adding a separate paragraph in the Introduction.
- Response: We tried to update of the manuscript and now explicitly addresses the main research question in Introduction.
- The authors do not provide adequate evidence regarding the novelty of the manuscript. In the revised version, the authors must elaborate in the Discussion about the novelty and the originality of the manuscript. They do not make the reader understand what specific gap in the field the paper address. This should be made clear in both the Introduction and in the Discussion.
- Response: Updated. The novelty and originality of the manuscript are elaborated upon in the Discussion section to clarify the specific gap addressed in the field.
- The authors do not elaborate on the value added by their manuscript in comparison to other published material. The authors must add a separate sub-section in the Discussion to present the additions in the literature brought forward by their study.
3.Response: Updated
- With regard to methodologies, the authors must specific justification for implementing the equations used in their study. Not all readers may be familiar with such complex equations, therefore the authors must explain in details why they used these equations and also they must justify their use. The authors should use confirmed negative control and confirmed positive controls in all their laboratory work and also they must present data about validation of their findings versus the confirmed negative and positive control material.
4.Response: Updated the Material and methods (equations are provided in box for clear understanding)
- The concluding remarks about canagliflozin and tepotinib have strong binding affinity with the viral proteins have not been fully documented in the manuscript and thus are not consistent with the results and the arguments in the Discussion. The authors should tone down their conclusions. Also the comment about the future performance of experimental studies should be fully justified, as it is not covered well in the Discussion.
5.Response: Thank you for the comment. We updated introduction and discussion section and future performance of experimental studies and limitations (Line No. 604-619). In the discussion, it is mentioned about the limitation of this study and need future studies to provide justification through experimental validation.
- Some recently published (January and February 2024) references are missing and can be added to improve the Discussion.
6.Response: Updated. (references: Yadav, D., Rao, G. S. N., Paliwal, D., Singh, A., Alam, A., Kumar Sharma, P., ... & Kumar, Y. Cracking the Code of Lumpy Skin Disease: Identifying Causes, Symptoms and Treatment Options for Livestock Farmers. Infectious Disorders-Drug Targets (Formerly Current Drug Targets-Infectious Disorders) 2024, 24(5), 57-71.)
Yousaf, M. A., Basheera, S., & Sivanandan, S. Inhibition of Monkeypox Virus DNA Polymerase Using Moringa oleifera Phytochemicals: Computational Studies of Drug-Likeness, Molecular Docking, Molecular Dynamics Simulation and Density Functional Theory. Indian Journal of Microbiology 2024, 1-18.
- Comments on figures: the visualisation of the manuscript is above average, but the authors must use similar colours to depict the same structure throughout the manuscript. -For the two figures of the protein structure, please add a new presentation of the structure from a different viewpoint to allow better understanding of the structure by readers.
7.Response: Thank you for mentioning the comment regarding colours of the structures Updated Figure for better understanding. For all the figures ligands and protein are carefully depicted the same colour. Only figure 2 is depicted in multiple colours to better visualize the conserved domain resembles with other DNA polymerase family.
-The authors should make clear that the study refers to an in silico project, as this does not become clear at the moment.
-The Introduction should be shorten, as it includes some well-established information.
Response: Updated (we mention focus of this study, aim and objectives Line no. 10-107s).
-The equation for the examination of the glide can be inserted within a box, for better presentation.
Response: Updated (put them in a box)
-Same for the equation for the energy for the protein ligand.
Response: Updated
-For the two figures of the protein structure, please add a new presentation of the structure from a different viewpoint to allow better understanding of the structure by readers.
Response: Updated
-The Discussion should be divided into three sub-sections for better flow of the reading.
Response: We tried to update the discussion section.
-Some recent references are missing and can be added to improve the Discussion.
Response: Updated (refer to comment 6)
-Please add a subsection in Discussion emphasizing the clinical significance of the findings.
Response: Updated (line No.584-588)
Overall: Acceptance after making the above changes.
Response: Thanks for your comments, suggestions and time and effort to improve the manuscript.
Reviewer 2 Report
Comments and Suggestions for Authors
The authors conducted a bioinformatic analysis of the DNA polymerase of the lumpy skin disease virus to search for potential inhibitors among drugs and phytocomponents. The authors put a lot of effort into writing the manuscript, but nevertheless there are a number of recommendations for improving it.
Major:
In the introduction, it is recommended to clearly formulate the purpose of the research (hypothesis). It is necessary to clearly separate studies on viral inhibition and DNA polymerase inhibition. Inhibition of the virus can occur at different stages of viral infection and have different mechanisms. In addition, it is recommended to exclude discussion of the presented results from the “Introduction” section.
It is also recommended to add information about the mechanisms of viral inhibition by drugs to the “Introduction” and “Discussion” sections
Regarding the use of ivermectin as a positive control. The mechanism of inhibition of the nodular dermatitis virus has not been studied for this drug. Toker et al., 2022, show that ivermectin inhibits lumpy skin disease virus at several stages of infection (attachment, entry, replication), although the mechanisms of action remain unclear.
Minor:
Line 67: Reference 14 is not used correctly
Lines 122-123: Editing required
Line 138: Supplementary data not relevant to this manuscript
Figures 6 and 7 merged into one.
Lines 570-573: It is not clear how Table 1 and Table S1 are related to this conclusion
Author Response
Reviewer 2
The authors conducted a bioinformatic analysis of the DNA polymerase of the lumpy skin disease virus to search for potential inhibitors among drugs and phytocomponents. The authors put a lot of effort into writing the manuscript, but nevertheless there are a number of recommendations for improving it.
Response: Thanks for the reviewer comments, critiques, and suggestions which help improve the present state of the manuscript. We appreciate your valuable comments and suggestions.
Major:
In the introduction, it is recommended to clearly formulate the purpose of the research (hypothesis). It is necessary to clearly separate studies on viral inhibition and DNA polymerase inhibition. Inhibition of the virus can occur at different stages of viral infection and have different mechanisms. In addition, it is recommended to exclude discussion of the presented results from the “Introduction” section.
Response: Thanks for the reviewer comments, critiques, and suggestions which help improve the present state of the manuscript. We appreciate your valuable comments and suggestions.
It is also recommended to add information about the mechanisms of viral inhibition by drugs to the “Introduction” and “Discussion” sections
Response: Introduction now clearly presents the purpose of the research and separates studies on viral inhibition and DNA polymerase inhibition. Discussions of the presented results have been excluded from the Introduction section. For example“Updated Line No. L590-617”
Regarding the use of ivermectin as a positive control. The mechanism of inhibition of the nodular dermatitis virus has not been studied for this drug. Toker et al., 2022, show that ivermectin inhibits lumpy skin disease virus at several stages of infection (attachment, entry, replication), although the mechanisms of action remain unclear.
Response: A brief discussion of the limitations of the current study has been included in the Discussion section, addressing the lack of experimental validation and the choice of ivermectin as a positive control.
mention separate paragraph in Discussion section (L590-617).
Minor:
Line 67: Reference 14 is not used correctly
Response: Updated
Lines 122-123: Editing required
Response: Updated
Line 138: Supplementary data not relevant to this manuscript
Response: attached correct one
Figures 6 and 7 merged into one.
Response: Updated (two figures (6 and 7) merged as 6 no figure)
Lines 570-573: It is not clear how Table 1 and Table S1 are related to this conclusion
Response: Updated

Reviewer 3 Report
Comments and Suggestions for Authors
The authors present an interesting study that aimed to utilise computational methods to identify potential compounds that have the capacity to bind DNA polymerase from lumpy skin disease virus (LSDV).
The introduction is quite long and some consideration should be given to shortening it. It also requires some modifications to ensure that the information provided is presented logically. For example, the paragraph (66-75) describing the emergence of the virus could be moved up to the second sentence.
The discussion is also very long, and efforts should be made to shorten it. Having said that the current version does not address the limitations of the current study. Key limitations that are not discussed are, firstly, that none of the proposed treatments have been experimentally validated. A brief discussion on this point with some relevant examples where similar approaches have been used to identify drugs that have progressed to experimental validation would be ideal. On this point throughout the manuscript the authors often use terms that suggest their interactions are absolutely certain, rather than predictions.
The second important limitation of the current study in my opinion is the selection of ivermectin as the “positive control”. While ivermectin has been shown to have the capacity to inhibit the lifecycle of LSDV, this effect may not be through inhibition of DNA polymerase. Recent studies on bovine herpesvirus 1 have elucidated some of the underpinning mechanisms of how ivermectin inhibits this virus. The studies suggest ivermectin interferes with the transport of viral proteins (including DNA pol) into the nucleus. Of course, this mode of action would not be effective against the poxviruses. However, published studies have identified potential inhibitors of monkey poxvirus DNA polymerase and perhaps one of these might be a more appropriate control compound.
Line 31 I am unsure how “poor bioavailability” relates to this point on vaccines. Bioavailability to me is more related to drug delivery, not vaccines. Please comment on how this term is relevant in this context.
Line 32 suggest revision “targeting the LSDV encoded RNA polymerase protein (gene LSDV039) for further investigation”
The LSDV DPol is encoded by the viral genome, to say it is “LSD-associated” suggest it is more related to the disease, LSD, rather than the pathogen.
Line 40 suggest revision “phytocompounds as potential LSD therapeutics.”
Line 40 suggest revision “close genetic relationships to sheep poxvirus (SSPV) and goat poxvirus (GTPV).”
I do not think “kinship” is the correct term in this context.
Line 60 suggest revision “exhibit clinical signs such”
Clinical signs is the correct term in this context.
Line 77 suggest revision “to stimulate immune responses oweing”
Line 81 I am not sure what the authors mean by the phrase “vaccinated animals remain ill”. Typically, vaccines are used to prevent disease, not to treat disease. Please review the text and modify as required to ensure the intended message is clear.
Line 86 Please add an appropriate citation for the quoted efficacy estimate in the Balkan region.
Line 98 Please review this sentence regarding the effectiveness of ivermectin against LSDV. It is unclear how there can be two estimates for both viral attachment and penetration.
Line 127 I would recommend replacing “effectively” with “potentially” given the current study does not test any compounds against LSDV.
Line 138 There is no supplemental Table S1 provided. The supplemental files that were attached do not appear to be related to this study.
Line 142 The date the provided weblinks were last accessed should be added here and elsewhere in the manuscript.
Line 164 Was this library created for the current study or a previous study? If it was generated for the current study, I do not see how there could be a reference for it.
Line 171 The legend should be below the image.
Line 290 The figure legend is not associated with the image, and appears on Line 311.
Line 291 suggest revision “Twelve of the 156 putative genes encoded by the LSDV genome were identified as potentially playing roles in viral DNA replication (Table S1) [6, 63].”
Line 292-294 Were any specific criteria applied to select DNA polymerase as the focus of this study? The point is that, from the outset of the study, DNA polymerase would be the logical target for drug development for LSDV.
Line 293 suggest revision “LSDV039 gene that encoded DNA polymerase enzyme”
Lines 295-299 – This text is more introductory (perhaps discussion) and is not suited to a results section. Suggest deletion.
Line 340 Is it possible to modify this image? Several of the amino acid residues either overlap with each other or other elements of the figure making them impossible to discern.
The legend should also describe the key features of the image to enable interpretation in the absence of the main text. For example;
What are the “yellow and red regions”?
What do the black squares represent?
What is the importance of the amino acids shown in red?
I thought these might be the binding residues, but there seems to be minimal overlap with the residues shown in Table 1.
What do the lowercase letters represent?
Line 346 What do the grey bars indicate?
Line 348 suggests revision “Potential antiviral compounds”
I think having “four” here is confusing, as it is the final result, whereas this text should describe the processes used in identifying the final four candidates.
Line 353 I would strongly recommend the authors include the outputs from these analyses for all compounds screened as supplemental tables. This would be important information for someone looking to replicate the study.
Line 364 suggest revision “energies for potential inhibitors of LSDV DNA polymerase.”
Line 475-501 This paragraph is mostly background and more suited to the introductory text. Given the introductory text is already long, this additional text could be deleted.
Comments on the Quality of English Language
See comments to authors.
Author Response
Reviewer 3
The authors present an interesting study that aimed to utilise computational methods to identify potential compounds that have the capacity to bind DNA polymerase from lumpy skin disease virus (LSDV).
The introduction is quite long and some consideration should be given to shortening it. It also requires some modifications to ensure that the information provided is presented logically. For example, the paragraph (66-75) describing the emergence of the virus could be moved up to the second sentence.
Response: Thanks for the reviewer comments, critiques, and suggestions which help improve the present state of the manuscript. The Introduction has been shortened and reorganized for logical presentation, with additional modifications for clarity.
(Updated)
The discussion is also very long, and efforts should be made to shorten it. Having said that the current version does not address the limitations of the current study. Key limitations that are not discussed are, firstly, that none of the proposed treatments have been experimentally validated. A brief discussion on this point with some relevant examples where similar approaches have been used to identify drugs that have progressed to experimental validation would be ideal. On this point throughout the manuscript the authors often use terms that suggest their interactions are absolutely certain, rather than predictions.
The second important limitation of the current study in my opinion is the selection of ivermectin as the “positive control”. While ivermectin has been shown to have the capacity to inhibit the lifecycle of LSDV, this effect may not be through inhibition of DNA polymerase. Recent studies on bovine herpesvirus 1 have elucidated some of the underpinning mechanisms of how ivermectin inhibits this virus. The studies suggest ivermectin interferes with the transport of viral proteins (including DNA pol) into the nucleus. Of course, this mode of action would not be effective against the poxviruses. However, published studies have identified potential inhibitors of monkey poxvirus DNA polymerase and perhaps one of these might be a more appropriate control compound.
Response: We mention a limitation para of this study discussion section. ( Line No. 601-617) mentioned two points (1). proposed treatments have been experimentally validated. (2) the selection of ivermectin as the “positive control” and it’s limitation.
Line 31 I am unsure how “poor bioavailability” relates to this point on vaccines. Bioavailability to me is more related to drug delivery, not vaccines. Please comment on how this term is relevant in this context.
Response: absolutely correct, we agree with your comments. (updated)
Line 32 suggest revision “targeting the LSDV encoded RNA polymerase protein (gene LSDV039) for further investigation”
Response: updated
The LSDV DPol is encoded by the viral genome, to say it is “LSD-associated” suggest it is more related to the disease, LSD, rather than the pathogen.
Response: updated
Line 40 suggest revision “phytocompounds as potential LSD therapeutics.”
Response: updated
Line 40 suggest revision “close genetic relationships to sheep poxvirus (SSPV) and goat poxvirus (GTPV).”
Response: updated
I do not think “kinship” is the correct term in this context.
Response: added “kinship”
Line 60 suggest revision “exhibit clinical signs such”
Response: Revised
Clinical signs is the correct term in this context.
Response: Revised
Line 77 suggest revision “to stimulate immune responses oweing”
Response: Revised
Line 81 I am not sure what the authors mean by the phrase “vaccinated animals remain ill”. Typically, vaccines are used to prevent disease, not to treat disease. Please review the text and modify as required to ensure the intended message is clear.
Response: Revised
Line 86 Please add an appropriate citation for the quoted efficacy estimate in the Balkan region.
Response: deleted this part
Line 98 Please review this sentence regarding the effectiveness of ivermectin against LSDV. It is unclear how there can be two estimates for both viral attachment and penetration.
Response: Updated
Line 127 I would recommend replacing “effectively” with “potentially” given the current study does not test any compounds against LSDV.
Response: Updated
Line 138 There is no supplemental Table S1 provided. The supplemental files that were attached do not appear to be related to this study.
Response: uploaded correct supplementary file
Line 142 The date the provided weblinks were last accessed should be added here and elsewhere in the manuscript.
Response: provided the link
Line 164 Was this library created for the current study or a previous study? If it was generated for the current study, I do not see how there could be a reference for it.
Response: library created for the current study ( provides supplementary files)
Line 171 The legend should be below the image.
Response: Updated
Line 290 The figure legend is not associated with the image, and appears on Line 311.
Response: Corrected
Line 291 suggest revision “Twelve of the 156 putative genes encoded by the LSDV genome were identified as potentially playing roles in viral DNA replication (Table S1) [6, 63].”
Response: revised
Line 292-294 Were any specific criteria applied to select DNA polymerase as the focus of this study? The point is that, from the outset of the study, DNA polymerase would be the logical target for drug development for LSDV.
Response “ Updated (L 293-297)
Line 293 suggest revision “LSDV039 gene that encoded DNA polymerase enzyme”
Response: revised
Lines 295-299 – This text is more introductory (perhaps discussion) and is not suited to a results section. Suggest deletion.
Response “ deleted”
Line 340 Is it possible to modify this image? Several of the amino acid residues either overlap with each other or other elements of the figure making them impossible to discern.
Response: Thanks for your observation. We tried to update and mention its overlapping amino acids name in figure legend. but it’s a software generated “ Updated
The legend should also describe the key features of the image to enable interpretation in the absence of the main text. For example;
Response: Updated (Line no. 358-368)
What are the “yellow and red regions”?
Response: Updated(Line no. 361-371)
What do the black squares represent?
Response: mentioned (Line no. 358-368)
What is the importance of the amino acids shown in red?
Response Updated (Line no. 358-368)
I thought these might be the binding residues, but there seems to be minimal overlap with the residues shown in Table 1.
Response: Updated
What do the lowercase letters represent?
Response: Updated
Line 346 What do the grey bars indicate?
Response Updated
Line 348 suggests revision “Potential antiviral compounds”
I think having “four” here is confusing, as it is the final result, whereas this text should describe the processes used in identifying the final four candidates.
Response Updated
Line 353 I would strongly recommend the authors include the outputs from these analyses for all compounds screened as supplemental tables. This would be important information for someone looking to replicate the study.
Response: provided as supplementary files (S2 and S3).
Line 364 suggest revision “energies for potential inhibitors of LSDV DNA polymerase.”
Response: Updated
Line 475-501 This paragraph is mostly background and more suited to the introductory text. Given the introductory text is already long, this additional text could be deleted.
Response: deleted

Round 2
Reviewer 2 Report
Comments and Suggestions for Authors The authors have done a great job of improving the manuscript based on the recommendations. Minor It is necessary to correctly place the legend to Figure 2